

# Exact mean-field solution of a spin chain with short-range and long-range interactions

Etienne Granet[*]

Kadanoff Center for Theoretical Physics, University of Chicago,
5640 South Ellis Ave, Chicago, IL 60637, USA

[*] egranet@uchicago.edu

## Abstract

We consider the transverse field Ising model with additional all-to-all interactions between the spins. We show that a mean-field treatment of this model becomes *exact* in the thermodynamic limit, despite the presence of 1D short-range interactions. Namely, we show that the eigenstates of the model are coherent states with an amplitude that varies through the Hilbert space, within which expectation values of local observables can be computed with mean-field theory. We study then the thermodynamics of the model and identify the different phases. Among its peculiar features, this 1D model possesses a second-order phase transition at finite temperature and exhibits inverse melting.

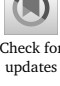

# 1 Introduction

Mean field theory (MFT) is a useful approximation that gives a rough qualitative idea of the behaviour of a model that is intractable otherwise, see e.g. [1]. But this approximation is rarely exact. Exception cases include models where each component interacts with a large number of other components, such as lattices with infinite dimensions [2–5], or long-range interacting models with a power-law exponent smaller than the dimension [6–9]. It also includes cases where, on the contrary, the components do not interact directly with their neighbours, such as infinite-temperature limits [10], or only interact collectively through a component that is singled out [11,12]. The common feature of these cases is that the eigenstates are tensor products at each site, and correlations are constant in space. But in presence of nearest neighbour interactions, MFT is expected to be only an approximation.

In this work, we consider the 1D Transverse Field Ising Model (TFIM) [13] with additional all-to-all interactions between the spins, scaled in a way that the all-to-all interactions and the nearest neighbour interactions in the TFIM both contribute to the energy densities in the thermodynamic limit. We show that for this model, MFT becomes exact in the thermodynamic limit, in the sense that the energy density of any state, as well as the expectation value of local operators, can be computed with a MFT Hamiltonian. This MFT Hamiltonian is the TFIM in which the transverse magnetic field is determined self-consistently, see below in Section 3.6. This includes in particular equilibrium expectation values of local operators at zero or finite temperature in the thermodynamic limit. Although all-to-all interactions are a usual feature of models for which MFT is exact, it is unexpected that the 1D nearest neighbour interactions

present in the TFIM do not spoil the exactness of MFT. In particular, the eigenstates of the model are *not* tensor products at each site, and display non-constant correlations in space. To establish this result, we show that one can obtain eigenstates of the model in terms of coherent states with an amplitude that varies through the Hilbert space. This consists in a generalization of the TFIM eigenstates which are regular coherent states, i.e. with an amplitude that is constant through the Hilbert space.

From this exact solution, we study then the behaviour of the model at both zero and finite temperature, and find a number of interesting features. We find that at zero temperature, the all-to-all interactions displace the critical value of the magnetic field, and marginally corrects the critical behaviour of observables. More notably, the Hamiltonian exhibits a second-order phase transition at finite temperature. This is allowed in this 1D model because of the long-range interactions and does not contradict Peierls's argument that applies to short-range interactions only. For the simplicity of its solution, this model could thus serve as a useful toy model for finite temperature transitions in quantum models, usually appearing in unsolvable 2D models. Finally, there is a tiny region in parameter space where the model presents inverse melting/freezing, i.e. a region where increasing the temperature drives the system into an ordered phase [15]. This seems to be the simplest quantum model with this behaviour [16–18].

Finally, let us mention occurrences of models with all-to-all interactions in the literature. The model we study has itself appeared in different contexts, such as particle-number-conserving version the Kitaev wire model [19], power-law interactions with exponent smaller than 1 [8, 20, 21], or in models of quantum optics at zero temperature [22]. Celebrated models with all-to-all interactions are the Curie-Weiss model [23] and the Lipkin-Meshkov-Glick model [24], and different models with all-to-all interactions have attracted attention recently [25–29].

*Note added.* After appearance of this manuscript on the arXiv, Ref [30] was brought to our attention, in which it was shown with a different approach that the energy density at thermal equilibrium can be computed with MFT in a class of models that contains (1).

## 2 Model and notations

### 2.1 The Hamiltonian

We consider the Hamiltonian on a chain with $L$ sites

$$H(h,\lambda) = -\sum_{j=1}^{L} \sigma_j^z \sigma_{j+1}^z + h \sum_{j=1}^{L} \sigma_j^x + \frac{\lambda}{L} \sum_{j,k=1}^{L} \sigma_j^x \sigma_k^x, \tag{1}$$

with $\sigma_j^{x,y,z}$ the Pauli matrices at site $j$, and $h, \lambda$ parameters. We impose periodic boundary conditions $\sigma_{L+1}^{x,y,z} = \sigma_1^{x,y,z}$. We note that the rightmost all-to-all interaction term has been rescaled by a factor $1/L$ in order to be of order $\mathcal{O}(L)$ as the other terms. The case $\lambda = 0$ corresponds to the Transverse Field Ising Model (TFIM), which is solvable in finite size $L$ [13, 31].

### 2.2 Notations

The Hamiltonian commutes with the symmetry operator

$$S = \prod_{j=1}^{L} \sigma_j^x, \tag{2}$$

and so splits into two sectors where $S = \pm 1$. We perform a Jordan-Wigner transformation by introducing operators $c_j$ that satisfy canonical anticommutation relations $\{c_j, c_k\} = 0$ and $\{c_j, c_k^\dagger\} = \delta_{j,k}$, and such that

$$\sigma_j^x = 1 - 2c_j^\dagger c_j, \qquad \sigma_j^z = (c_j + c_j^\dagger)\prod_{\ell=1}^{j-1}(1 - 2c_\ell^\dagger c_\ell). \tag{3}$$

The fermions are given periodic boundary conditions $c_{L+1} = c_1$ in the $S = -1$ sector and antiperiodic boundary conditions $c_{L+1} = -c_1$ in the $S = 1$ sector [13, 31]. We then define

$$c(k) = \frac{1}{\sqrt{L}}\sum_{j=1}^{L} e^{ijk}c_j, \tag{4}$$

with in the $S = 1$ sector

$$k \in K = \left\{\frac{2\pi(n+1/2)}{L}, n = -L, \ldots, L-1\right\}, \tag{5}$$

and in the $S = -1$ sector

$$k \in K = \left\{\frac{2\pi n}{L}, n = -L, \ldots, L-1\right\}. \tag{6}$$

For each sector, given $\boldsymbol{k} \subset K$ a subset of momenta with an even/odd number of elements for $S = \pm 1$, we define the state

$$|\boldsymbol{k}\rangle = \prod_{k \in \boldsymbol{k}} c_k^\dagger |0\rangle, \tag{7}$$

with $|0\rangle$ the tensor product of $+1$ eigenstates of $\sigma_j^x$ at each site. In this expression, an arbitrary fixed ordering of the $c_k^\dagger$'s is chosen for each $\boldsymbol{k}$. We choose this ordering such that

$$|\boldsymbol{k} \cup \{q, -q\}\rangle = c_{-q}^\dagger c_q^\dagger |\boldsymbol{k}\rangle. \tag{8}$$

Finally, we define the shorthand notations

$$S_{zz} = \sum_{j=1}^{L} \sigma_j^z \sigma_{j+1}^z, \qquad S_x = \sum_{j=1}^{L} \sigma_j^x, \tag{9}$$

in terms of which the Hamiltonian is

$$H(h, \lambda) = -S_{zz} + hS_x + \frac{\lambda}{L}S_x^2. \tag{10}$$

For future reference, let us give the matrix elements of $S_{zz}$ and $S_x$ in the basis of the $|\boldsymbol{k}\rangle$'s. Those of $S_x$ are

$$\langle \boldsymbol{k}|S_x|\boldsymbol{k}\rangle = L - 2\sum_{k \in \boldsymbol{k}} 1, \tag{11}$$

and all the other matrix elements are zero. As for $S_{zz}$, we have for $q \in K$

$$\langle \boldsymbol{k}|S_{zz}|\boldsymbol{k}\rangle = 2\sum_{k \in \boldsymbol{k}} \cos k, \qquad \langle \boldsymbol{k}|S_{zz}|\boldsymbol{k} \cup \{q, -q\}\rangle = -2i\sin q, \tag{12}$$

if $q, -q \notin \boldsymbol{k}$. All the other non-related matrix elements are zero.

## 2.3 Relation with other models

In this Section we review models that can be mapped to $H(h, \lambda)$.

### 2.3.1 Particle-number-conserving Kitaev model

In [19] was introduced particle-number conserving version of the Kitaev wire model [32]. The Hamiltonian reads

$$H_{SC} = -\frac{t}{2}\sum_{j=1}^{L-1}(c_j^\dagger c_{j+1} + c_{j+1}^\dagger c_j) - \frac{\Delta}{2}\sum_{j=1}^{L-1}(c_j c_{j+1} e^{i\phi} + c_{j+1}^\dagger c_j^\dagger e^{-i\phi}) - \mu N_w + \frac{4\mathcal{E}_c}{L}(n - n_c)^2, \quad (13)$$

where $t, \Delta, \mu, n_c, \mathcal{E}_c$ are real parameters, $c_j$ canonical fermions, $N_w = \sum_{j=1}^{L} c_j^\dagger c_j$ is the operator counting the number of particles in the wire, $n$ the operator counting the number of Cooper pairs and $e^{i\phi}$ is a ladder operator for $n$, i.e. $[n, e^{i\phi}] = e^{i\phi}$. Particle number conservation in the wire and superconductor is implemented by requiring $N = N_w + 2n$ to be fixed constant. The scaling with $L$ of the charging energy $\frac{\mathcal{E}_c}{L}$ of the superconductor comes from the Coulomb interaction in the 3D superconductor [19]. At $t = \Delta$, by writing $n$ in terms of $N_w$ and using the Jordan-Wigner transformation, it is seen that up to an additive constant this model is equivalent to $\frac{t}{2}H(h, \lambda)$ in the thermodynamic limit with

$$h = \frac{\mu}{t} - \frac{2\mathcal{E}_c}{Lt}(2n_c + 2L - N), \qquad \lambda = \frac{2\mathcal{E}_c}{t}. \quad (14)$$

In [19, 33] were established multiple properties that the Hamiltonian $H_{SC}$ shares with the Hamiltonian obtained by treating the quadratic term $(n - n_c)^2$ in a mean-field way.

### 2.3.2 Many-body cavity systems

Similar systems to (1) can be realized with cold atoms on optical lattices interacting with a cavity, see e.g. [34–39]. All-to-all interactions between spins can appear as a resulting coupling to an external ancilla degree of freedom. One example is the so-called Dicke-Ising Hamiltonian. The Dicke Hamiltonian is a fundamental model for light-matter interactions that reads [40]

$$H_{\text{Dicke}} = \omega_c a^\dagger a + \frac{g}{\sqrt{L}}(a + a^\dagger)\sum_{j=1}^{L} \sigma_j^x, \quad (15)$$

with a bosonic operator $a$ satisfying the canonical commutation relation $[a, a^\dagger] = 1$, and with $g, \omega_c > 0$ real parameters. The Dicke-Ising model is then obtained by imposing Ising interactions between the spins

$$H_{\text{Dicke-Ising}} = H_{\text{Dicke}} - \sum_{j=1}^{L} \sigma_j^z \sigma_{j+1}^z + h\sum_{j=1}^{L} \sigma_j^x. \quad (16)$$

It was shown in [22] that the ground state energy density of the Dick-Ising model is the same as the ground state of $H(h, \lambda)$ for

$$\lambda = -\frac{g^2}{4\omega_c}. \quad (17)$$

### 2.3.3 Long-range Ising chain with Kac rescaling

The Hamiltonian (1) is also related to a long-range Ising chain. Let us define

$$H = -\sum_{j=1}^{L} \sigma_j^z \sigma_{j+1}^z + h \sum_{j=1}^{L} \sigma_j^x + \mu \sum_{i \neq j} \frac{\sigma_i^x \sigma_j^x}{|i-j|_L^\alpha}, \tag{18}$$

for some parameters $\mu$ and $\alpha$. We set here $|n|_L = \min(|n|, |n+L/2|, |n-L/2|)$ in the last term to be compatible with the periodic boundary conditions. If $\alpha > 1$, this last term is of order $\mathcal{O}(L)$. However, if $\alpha < 1$ it is of order $\mathcal{O}(L^{2-\alpha})$. In particular, it makes the ground state energy super-extensive. A simple way to define a long-range interacting model with $\alpha < 1$ with extensive energy is to consider the so-called Kac prescription [41], that is setting

$$\mu = \frac{\nu}{L^{1-\alpha}}, \tag{19}$$

for $\nu$ constant. This ensures that the energy levels of $H$ are of order $\mathcal{O}(L)$. But then, one sees that any term with $|i-j| = \mathcal{O}(1)$ is suppressed in the thermodynamic limit. Let us consider $|\psi\rangle$ a state that satisfies clustering for $\sigma^x$, namely such that for large $|i-j|$

$$\langle\psi|\sigma_i^x \sigma_j^x|\psi\rangle = \langle\psi|\sigma_i^x|\psi\rangle\langle\psi|\sigma_j^x|\psi\rangle + o(|i-j|^0). \tag{20}$$

Then $|\psi\rangle$ has the same energy density in $H$ and in $H(h, \lambda)$

$$\frac{1}{L}\langle\psi|H|\psi\rangle = \frac{1}{L}\langle\psi|H(h, \lambda)|\psi\rangle + o(L^0), \tag{21}$$

with

$$\lambda = \frac{2^\alpha}{1-\alpha}\nu. \tag{22}$$

Hence, provided the clustering property holds for an eigenstate of $H$, this wave function is also an eigenstate of the all-to-all interacting chain $H(h, \lambda)$ in the thermodynamic limit. This kind of reduction from a power-law interacting model with exponent smaller than 1 to an all-to-all interacting model has been observed and shown for the ground state of other models [8, 20, 21].

## 3 Diagonalizing $H(h, \lambda)$ in the thermodynamic limit

### 3.1 Preservation of pair structure

In the remainder of the paper, we will fix the sector to $S = 1$, and so set $K$ to (5). Let us first show a block diagonal structure of the Hamiltonian $H(h, \lambda)$. From the matrix elements (11) and (12), we see that $H(h, \lambda)$ can have non-zero matrix elements in the basis of the $|\boldsymbol{k}\rangle$'s only between states that differ by a pair of momenta $q, -q$. This suggests to decompose a set of momenta $\boldsymbol{k} \subset K$ into momenta $k \in \boldsymbol{k}$ that are paired, i.e. for which $-k \in \boldsymbol{k}$, and momenta that are single, for which $-k \notin \boldsymbol{k}$. Specifically, we introduce

$$K_+ = \{k \in K \mid k > 0\}, \tag{23}$$

the set of positive momenta, and for $\boldsymbol{k} \subset K_+$ define $\bar{\boldsymbol{k}} \subset K$ as

$$\bar{\boldsymbol{k}} = \boldsymbol{k} \cup (-\boldsymbol{k}). \tag{24}$$

Besides, we call $\boldsymbol{s} \subset K$ a set of single momenta if it has an even number of elements and if $-s \notin \boldsymbol{s}$ for $s \in \boldsymbol{s}$. Then we define

$$K_+^{\boldsymbol{s}} = \{k \in K \quad | \quad k > 0, \ k \notin \boldsymbol{s}, \ -k \notin \boldsymbol{s}\} , \tag{25}$$

the set of strictly positive momenta that do not belong to $\boldsymbol{s}$ and whose opposite do not belong to $\boldsymbol{s}$. Any set of momenta $\boldsymbol{p} \subset K$ can be decomposed uniquely as $\boldsymbol{p} = \bar{\boldsymbol{k}} \cup \boldsymbol{s}$ for some set of single momenta $\boldsymbol{s}$ and $\boldsymbol{k} \subset K_+^{\boldsymbol{s}}$. Hence, from the matrix elements (11) and (12), fixing a set of single momenta $\boldsymbol{s}$, we have that

$$H(h, \lambda) |\bar{\boldsymbol{k}} \cup \boldsymbol{s}\rangle \tag{26}$$

is a linear combination of states $|\bar{\boldsymbol{p}} \cup \boldsymbol{s}\rangle$ with $\boldsymbol{p} \subset K_+^{\boldsymbol{s}}$. Namely, $H(h, \lambda)$ splits into sectors with fixed set of single momenta $\boldsymbol{s}$, and so can be diagonalized separately in each.

For simplicity and lightness of the notations, we will present in details the diagonalization of $H(h, \lambda)$ in the sector where the single set of momenta is the empty set $\boldsymbol{s} = \emptyset$. The results for a generic set of single momenta $\boldsymbol{s}$ will then be given in Section 3.10.

## 3.2 Warm-up: solving the TFIM with coherent states

As a warm-up to the next sections, we show how to solve the TFIM case $\lambda = 0$ using the so-called coherent states. The *coherent state* $|\phi\rangle$ for a given function $\phi(q)$ defined on $K_+$, is the state defined by

$$|\phi\rangle = A \sum_{\boldsymbol{k} \subset K_+} \left( \prod_{k \in \boldsymbol{k}} i\phi(k) \right) |\bar{\boldsymbol{k}}\rangle = A \prod_{q \in K_+} \left( 1 + i\phi(q) c_{-q}^{\dagger} c_q^{\dagger} \right) |0\rangle , \tag{27}$$

with $A = \prod_{\in K_+} \frac{1}{\sqrt{1 + |\phi(k)|^2}}$ a normalization factor. The relevance of these states for the TFIM were first noticed in [42, 43]. It satisfies the "factorization property" for $q \notin \boldsymbol{k} \subset K_+$

$$\langle \bar{\boldsymbol{k}} \cup \{q, -q\} | \phi\rangle = i\phi(q) \langle \bar{\boldsymbol{k}} | \phi\rangle . \tag{28}$$

Let us look for an eigenstate of $H(h, 0)$ with $\lambda = 0$ under the form of a coherent state $|\phi\rangle$. Given the matrix elements (11) and (12) we have

$$\langle \bar{\boldsymbol{k}} | S_x | \phi\rangle = \left( L - 2 \sum_{k \in \boldsymbol{k}} 1 \right) \langle \bar{\boldsymbol{k}} | \phi\rangle , \tag{29}$$

and

$$\begin{aligned}
\langle \bar{\boldsymbol{k}} | S_{zz} | \phi\rangle &= \sum_{\boldsymbol{q} \subset K} \langle \bar{\boldsymbol{k}} | S_{zz} | \boldsymbol{q}\rangle \langle \boldsymbol{q} | \phi\rangle \\
&= \langle \bar{\boldsymbol{k}} | S_{zz} | \bar{\boldsymbol{k}}\rangle \langle \bar{\boldsymbol{k}} | \phi\rangle + \sum_{q \in \boldsymbol{k}} \langle \bar{\boldsymbol{k}} | S_{zz} | \bar{\boldsymbol{k}} \setminus \{q, -q\}\rangle \langle \bar{\boldsymbol{k}} \setminus \{q, -q\} | \phi\rangle \\
&\quad + \sum_{q \notin \boldsymbol{k}} \langle \bar{\boldsymbol{k}} | S_{zz} | \bar{\boldsymbol{k}} \cup \{q, -q\}\rangle \langle \bar{\boldsymbol{k}} \cup \{q, -q\} | \phi\rangle .
\end{aligned} \tag{30}$$

Hence, using the factorization property (28), we obtain

$$\langle \bar{\boldsymbol{k}} | H(h, 0) | \phi\rangle = E(\boldsymbol{k}) \langle \bar{\boldsymbol{k}} | \phi\rangle , \tag{31}$$

with the "candidate energy"

$$E(\boldsymbol{k}) = hL - 2 \sum_{k \in K_+} \phi(k) \sin k + 2 \sum_{q \in \boldsymbol{k}} \left[ -2h - 2\cos q + \left( \phi(q) - \tfrac{1}{\phi(q)} \right) \sin q \right] . \tag{32}$$

Since $H(h,0)$ preserves the pair structure, $|\phi\rangle$ is an eigenstate of $H(h,0)$ if and only if $E(\boldsymbol{k})$ is independent of $\boldsymbol{k}$. This is is equivalent to having for all $q \in K_+$

$$-2h - 2\cos q + (\phi(q) - \tfrac{1}{\phi(q)})\sin q = 0. \tag{33}$$

For each $q \in K_+$ there are two solutions for $\phi(q)$

$$\phi(q) = \frac{h + \cos q}{\sin q} \pm \sqrt{\left(\frac{h + \cos q}{\sin q}\right)^2 + 1}. \tag{34}$$

Each of these $2^{L/2}$ choices for the function $\phi$ gives an eigenstate of $H(h,\lambda)$

$$H(h,\lambda)|\phi\rangle = E|\phi\rangle, \tag{35}$$

with energy

$$E = hL - 2\sum_{k \in K_+} \phi(k)\sin k. \tag{36}$$

Let us denote $E_0$ the energy obtained by choosing the $+$ sign in (34) for all $k$. Denoting then $\boldsymbol{q} \subset K_+$ the subset of momenta for which the $-$ sign is chosen in (55), the energy of this eigenstate is

$$E = E_0 + 4\sum_{q \in \boldsymbol{q}} \sqrt{1 + h^2 + 2h\cos q}. \tag{37}$$

These are exactly the energies of the $2^{L/2}$ paired eigenstates of the TFIM Hamiltonian $H(h,0)$ [13, 31].

## 3.3 Density-resolved coherent state

Let us now generalize the approach of Section 3.2 to the non-integrable case $\lambda \neq 0$. We consider $|\phi\rangle$ a linear combination of paired states $|\bar{\boldsymbol{k}}\rangle$ with $\boldsymbol{k} \subset K_+$, and define $\phi_{\boldsymbol{k}}(q)$ with $q \in K_+$ by

$$\langle \bar{\boldsymbol{k}} \cup \{q,-q\}|\phi\rangle = i\phi_{\boldsymbol{k}}(q)\langle \bar{\boldsymbol{k}}|\phi\rangle. \tag{38}$$

We note that for any state with non-zero overlaps over the paired states $|\bar{\boldsymbol{k}}\rangle$ there exists such a function $\phi_{\boldsymbol{k}}(q)$ without further assumptions. Repeating the steps of Section 3.2, we obtain similarly

$$\langle \bar{\boldsymbol{k}}|H(h,\lambda)|\phi\rangle = E(\boldsymbol{k})\langle \bar{\boldsymbol{k}}|\phi\rangle, \tag{39}$$

with now the candidate energy

$$E(\boldsymbol{k}) = hL - 2\sum_{k \in K_+} \phi_{\boldsymbol{k}}(k)\sin k + 2\sum_{q \in \boldsymbol{k}}\left[-2h - 2\cos q + \left(\phi_{\boldsymbol{k}}(q) - \tfrac{1}{\phi_{\boldsymbol{k}\backslash q}(q)}\right)\sin q\right]$$
$$+ \frac{\lambda}{L}\left(L - 4\sum_{q \in \boldsymbol{k}} 1\right)^2. \tag{40}$$

The condition for $|\phi\rangle$ to be an eigenstate of $H(h,\lambda)$ is again that $E(\boldsymbol{k})$ is independent of $\boldsymbol{k}$. Contrary to the integrable case $\lambda = 0$, we see however that a coherent state, i.e. a state for which $\phi_{\boldsymbol{k}}(q) = \phi(q)$ is independent of $\boldsymbol{k}$, cannot satisfy this condition. Indeed, sets with only one element $\boldsymbol{k} = \{q\}$ for $q \in K_+$ would fix the value of $\phi(q) - \frac{1}{\phi(q)}$ through a condition similar to (33). But then, considering sets with two elements $\boldsymbol{k} = \{q_1, q_2\}$, we would necessarily have $E(\{q_1, q_2\}) \neq E(\{q_1\})$ if $\lambda \neq 0$.

To go further, we are going to consider an ansatz state $|\phi\rangle$ for which in the thermodynamic limit, $\phi_{\boldsymbol{k}}$ depends only on the density $\rho(k)$ of momenta in $\boldsymbol{k}$, denoted then $\phi_\rho$, and that

moreover $\phi_\rho$ has a smooth dependence in $\rho$. We are going to show that one can build indeed a functional $\phi_\rho$ that will make the candidate energy $E(\mathbf{k})$ independent of $\mathbf{k}$. So the state $|\phi\rangle$ is assumed to satisfy the factorization property in the thermodynamic limit

$$\langle \bar{\mathbf{k}} \cup \{q, -q\} | \phi \rangle = i\phi_\rho(q) \langle \bar{\mathbf{k}} | \phi \rangle , \tag{41}$$

with $\rho$ the density of $\mathbf{k}$. Such a state will be called *density-resolved coherent state*, in the sense that the function $\phi(q)$ involved in the coherent state factorization property (38) now depends on the density of momenta $\rho$ in the state for which the overlap is computed.

The functional $\phi_\rho(q)$ entering the definition of the density-resolved coherent state (41) cannot be chosen freely. The definition of $\phi_\mathbf{k}$ in (38) actually implies an important constraint on $\phi_\rho$ in the thermodynamic limit. Indeed, one can form a state by adding momenta in a different order, so that some consistency equations have to be satisfied on $\phi_\mathbf{k}$. For example, adding momenta $q_1$ or $q_2$ first implies

$$\phi_\mathbf{k}(q_1) \phi_{\mathbf{k} \cup q_1}(q_2) = \phi_\mathbf{k}(q_2) \phi_{\mathbf{k} \cup q_2}(q_1) . \tag{42}$$

We show in Appendix A that in the thermodynamic limit, the only resulting constraint on $\phi_\rho$ is

$$\frac{\partial_{\rho(q)} \phi_\rho(k)}{\phi_\rho(k)} = \frac{\partial_{\rho(k)} \phi_\rho(q)}{\phi_\rho(q)} , \tag{43}$$

for all $k, q \in K_+$.

Returning to (40), the candidate energy density $e(\rho) \equiv \frac{E(\mathbf{k})}{L}$ takes the following form in the thermodynamic limit

$$
\begin{aligned}
e(\rho) =& h - \frac{1}{\pi} \int_0^\pi \phi_\rho(k) \sin k \, dk + 2 \int_0^\pi \rho(k) \left[ -2h - 2\cos k + \left( \phi_\rho(k) - \frac{1}{\phi_\rho(k)} \right) \sin k \right] dk \\
& + \lambda \left( 1 - 4 \int_0^\pi \rho(k) dk \right)^2 .
\end{aligned}
\tag{44}
$$

For $|\phi\rangle$ to be an eigenstate of $H(h, \lambda)$ one requires that $e(\rho)$ is independent of $\rho$. Without constraints on $\phi_\rho$, one could make $e(\rho)$ independent of $\rho$ in many ways, e.g. by replacing the magnetic field $h$ in (34) by $h + 2\lambda(1 - 4\int_0^\pi \rho(k)dk)$. But the resulting $\phi_\rho$ would violate the constraint (43). The condition that $e(\rho)$ is independent of $\rho$ can be formulated as $\partial_{\rho(q)} e(\rho) = 0$ for all $q \in K_+$ and for all $\rho$. This is

$$
\begin{aligned}
\partial_{\rho(q)} e(\rho) =& 2 \left[ -2h - 2\cos q + \left( \phi_\rho(q) - \frac{1}{\phi_\rho(q)} \right) \sin q \right] - 8\lambda \left( 1 - 4 \int_0^\pi \rho(k) dk \right) \\
& + 2 \int_0^\pi \partial_{\rho(q)} \phi_\rho(k) \sin k \left[ \rho(k) \frac{1 + \phi_\rho(k)^2}{\phi_\rho(k)^2} - \frac{1}{2\pi} \right] dk = 0 .
\end{aligned}
\tag{45}
$$

However, finding a solution $\phi_\rho(k)$ to this equation that satisfies the constraint (43) is a priori a difficult task.

## 3.4 Dominant density

To go further, let us come back to the factorization property (41). Because of this relation, one sees that the overlap of $|\phi\rangle$ with a basis state $|\bar{\mathbf{k}}\rangle$ will be exponential in $L$ and take the following form in the thermodynamic limit

$$|\langle \bar{\mathbf{k}} | \phi \rangle|^2 = a[\rho] e^{Lb[\rho]} , \tag{46}$$

with $a, b$ some functionals of $\rho$ of order $L^0$. Here, (41) implies

$$e^{\partial_{\rho(q)} b[\rho]} = |\phi_\rho(q)|^2 \,. \tag{47}$$

Let us now consider $\mathcal{O}$ an observable that is local in the basis of the $|\boldsymbol{k}\rangle$'s, i.e. that has non-zero matrix elements $\langle \boldsymbol{k}|\mathcal{O}|\boldsymbol{q}\rangle \neq 0$ only if $\boldsymbol{q}$ has less than $n$ momenta that differ from $\boldsymbol{k}$, with $n$ fixed. We have then

$$\langle \phi |\mathcal{O}|\phi\rangle = \sum_{\boldsymbol{k},\boldsymbol{q} \subset K} \langle \boldsymbol{k}|\phi\rangle \langle \boldsymbol{q}|\phi\rangle^* \langle \boldsymbol{q}|\mathcal{O}|\boldsymbol{k}\rangle \,. \tag{48}$$

Using that $\mathcal{O}$ is local in the basis, one can write

$$\langle \phi |\mathcal{O}|\phi\rangle = \sum_{\boldsymbol{k} \subset K_+} |\langle \bar{\boldsymbol{k}}|\phi\rangle|^2 \left( \sum_{q_1,\dots,q_n} \prod_{j=1}^{n} i\tilde{\phi}_\rho(q_j)\langle \bar{\boldsymbol{q}}|\mathcal{O}|\bar{\boldsymbol{k}}\rangle \right), \tag{49}$$

where $\tilde{\phi}_\rho(q_j) = \phi_\rho(q_j)$ if $q_j \notin \boldsymbol{k}$ and $\tilde{\phi}_\rho(q_j) = -1/\phi_\rho(q_j)$ if $q_j \in \boldsymbol{k}$, with $\rho$ is the density of $\boldsymbol{k}$. The quantity in parentheses is of order at most polynomial in $L$, whereas the overlap $|\langle \bar{\boldsymbol{k}}|\phi\rangle|^2$ is exponential in $L$. Moreover, there are $e^{LS[\rho]}$ terms whose $\boldsymbol{k}$ has density $\rho$ in the thermodynamic limit, with the entropy

$$S[\rho] = -\frac{1}{2\pi} \int_0^\pi 2\pi\rho \log(2\pi\rho) + (1 - 2\pi\rho)\log(1 - 2\pi\rho) \,. \tag{50}$$

Hence in the thermodynamic limit, states $|\bar{\boldsymbol{k}}\rangle$ with density $\rho$ contribute to $\langle \phi |\mathcal{O}|\phi\rangle$ with a weight $e^{Lw[\rho]}$ where

$$w[\rho] = b[\rho] + S[\rho] \,. \tag{51}$$

It follows that in the thermodynamic limit, the states with density $\rho_*$ that maximises the exponent $w[\rho]$ will dominate exponentially. A necessary condition is $\partial_{\rho(q)} w[\rho] = 0$ at $\rho = \rho_*$, which gives

$$\partial_{\rho(q)} b[\rho_*] - \log \frac{\rho_*(q)}{\frac{1}{2\pi} - \rho_*(q)} = 0 \,, \tag{52}$$

namely

$$\rho_*(q) = \frac{1}{2\pi} \frac{|\phi_{\rho_*}(q)|^2}{1 + |\phi_{\rho_*}(q)|^2} \,. \tag{53}$$

Hence, as long as operators local in the basis are concerned, expectation values within a density-resolved coherent state $|\phi\rangle$ will depend *only* on the value of $\phi_{\rho_*}(q)$ where the dominant density $\rho_*$ satisfies (53).

We note that this condition (53) would also be satisfied by a local minimum of $w[\rho]$, whereas the dominant density $\rho_*$ should be a maximum. We will come back to this in Section 3.8.

## 3.5 The energy densities in the thermodynamic limit

Let us now come back to the condition for which $|\phi\rangle$ is an eigenstate of $H(h, \lambda)$ in the thermodynamic limit, i.e. $\partial_{\rho(q)} e(\rho) = 0$ written in (45). One sees that, remarkably, if $\phi_\rho(k)$ is real, this condition simplifies greatly at the dominant density $\rho_*$ since the coefficient in front of $\partial_{\rho(q)} \phi_\rho(k)$ vanishes. This yields the following quadratic equation on $\phi_{\rho_*}(k)$

$$-2h - 2\cos q + \left( \phi_{\rho_*}(q) - \frac{1}{\phi_{\rho_*}(q)} \right) \sin q - 4\lambda \left( 1 - 4 \int_0^\pi \rho_*(k) \mathrm{d}k \right) = 0 \,. \tag{54}$$

We obtain thus the exact expression

$$\phi_{\rho_*}(q) = \frac{h + 2\lambda(1-4\mathcal{D}) + \cos q}{\sin q} \pm \sqrt{\left(\frac{h + 2\lambda(1-4\mathcal{D}) + \cos q}{\sin q}\right)^2 + 1}, \tag{55}$$

where we introduced

$$\mathcal{D} = \int_0^\pi \rho_*(k)\mathrm{d}k. \tag{56}$$

Hence, notably, we can determine the only needed value of $\phi_\rho$ in the thermodynamic limit without solving the full equation (45). The choice of the $\pm$ sign in (55) parametrizes different eigenstates, as in the TFIM case. For each of these choices, the condition on the dominant density (53) implies then that $\mathcal{D}$ satisfies the self-consistent equation

$$\mathcal{D} = \frac{1}{2\pi} \int_0^\pi \frac{\phi_{\rho_*}(q)^2}{1 + \phi_{\rho_*}(q)^2}\mathrm{d}k. \tag{57}$$

The value of the energy density of this eigenstate is then given by $e(\rho_*)$ in (44). This reads

$$e(\rho_*) = h + \lambda - \frac{1}{\pi} \int_0^\pi \phi_{\rho_*}(k) \sin k \mathrm{d}k - 16\lambda\mathcal{D}^2. \tag{58}$$

## 3.6 Interpretation in terms of a mean-field Hamiltonian

We can reformulate the previous results in a more familiar way. Let us denote

$$x = h + 2\lambda(1-4\mathcal{D}), \tag{59}$$

and $\phi_\pm(q)$ the expression (55) on the right-hand side. Introducing

$$\varepsilon_x(k) = \sqrt{1 + x^2 + 2x\cos k}, \tag{60}$$

we compute

$$\begin{aligned}
\phi_+(q)\sin q - \phi_-(q)\sin q &= 2\varepsilon_x(q), \\
\frac{\phi_+^2(q)}{1 + \phi_+^2(q)} - \frac{\phi_-^2(q)}{1 + \phi_-^2(q)} &= \partial_x \varepsilon_x(q).
\end{aligned} \tag{61}$$

Hence, introducing the density of excitations $0 \le \nu(k) \le \frac{1}{2\pi}$ indicating the density of the $k$'s for which the $-$ sign is chosen in (55), we find that the energy density $\mathcal{E}(\nu)$ reads

$$\mathcal{E}(\nu) = -\frac{1}{\pi} \int_0^\pi \varepsilon_x(k)(1 - 4\pi\nu(k))\mathrm{d}k - \frac{(x-h)^2}{4\lambda}, \tag{62}$$

where from (57) $x$ satisfies the equation

$$x + \frac{2\lambda}{\pi} \int_0^\pi \partial_x \varepsilon_x(k)(1 - 4\pi\nu(k))\mathrm{d}k = h. \tag{63}$$

We note that this equation is exactly the extremality condition of $\mathcal{E}(\nu)$ with respect to $x$, namely $\partial_x \mathcal{E}(\nu) = 0$. Such formulation precisely corresponds to a mean field TFIM with an effective transverse field $x$

$$H = -\sum_{j=1}^L \sigma_j^z \sigma_{j+1}^z + x \sum_{j=1}^L \sigma_j^x - \frac{(x-h)^2}{4\lambda}, \qquad x = h + 2\lambda\left\langle \frac{1}{L}\sum_{j=1}^L \sigma_j^x \right\rangle, \tag{64}$$

where the expectation value is taken in the eigenstate considered with fixed excitations $\nu(k)$. This amounts to making the mean-field "approximation" in $H(h, \lambda)$

$$\left(\sum_{j=1}^{L} \sigma_j^x\right)^2 \longrightarrow 2\sum_{j=1}^{L} \sigma_j^x \left\langle \sum_{j=1}^{L} \sigma_j^x \right\rangle - \left\langle \sum_{j=1}^{L} \sigma_j^x \right\rangle^2. \tag{65}$$

The derivation of the previous subsections of Section 3 shows that the MFT Hamiltonian (64) becomes *exact* in the thermodynamic limit.

### 3.7 Local correlations in the thermodynamic limit

The fact that eigenstates of $H(h, \lambda)$ are density-resolved coherent states also allows for an exact expression of a large class of expectation values and correlation functions. Indeed, for any observable $\mathcal{O}$ that is local in the basis of $k$'s, the reasoning of Section 3.4 applies and in the thermodynamic limit the expectation value of $\mathcal{O}$ is

$$\lim_{L \to \infty} \langle \phi_\rho | \mathcal{O} | \phi_\rho \rangle = \langle \phi_{\rho_*} | \mathcal{O} | \phi_{\rho_*} \rangle. \tag{66}$$

This means it can be computed within a (not density-resolved) coherent state with amplitude $\phi_{\rho_*}(k)$. For these states, techniques allow for the computation of any expectation value of operators that are local in the basis. We refer the reader to Ref [43] where expectation values of various observables are computed within a coherent state.

### 3.8 Computing $\phi_\rho$ for $\rho \neq \rho_*$

In this Section, we investigate the construction of a solution $\phi_\rho$ to (45) that satisfies (43), perturbatively in

$$\delta\rho(k) = \rho(k) - \rho_*(k). \tag{67}$$

To that end, we introduce $F_\rho(k)$ the functional defined by

$$\phi_\rho(k) = \phi_{\rho_*}(k) \exp F_\rho(k). \tag{68}$$

The constraint (43) translates into the fact that the quantity

$$f_n(q_1, \ldots, q_n) = \partial_{\rho(q_1)} \ldots \partial_{\rho(q_{n-1})} F_\rho(q_n)|_{\rho=\rho_*} \tag{69}$$

must be symmetric in its arguments $q_1, \ldots, q_n$ for all $q_i$'s and $n$. We recall that $|\phi\rangle$ is an eigenstate of $H(h, \lambda)$ if and only if the candidate energy density $e(\rho)$ in (44) is independent of $\rho$. We saw that the equation $\partial_\rho(q)e(\rho) = 0$ at $\rho = \rho_*$ yields an equation on $\phi_{\rho_*}$. Similarly, an equation on $f_n$ is obtained from

$$\partial_{\rho(q_1)} \ldots \partial_{\rho(q_n)} e(\rho)|_{\rho=\rho_*} = 0. \tag{70}$$

From the expression (44) for $e(\rho)$, and using the condition for the dominant density (53), we obtain for $n = 2$ the following algebraic Riccati equation on $f_2$

$$16\lambda + f_2(k,q)\left(\sin k\left(\phi_{\rho_*}(k) + \frac{1}{\phi_{\rho_*}(k)}\right) + \sin q\left(\phi_{\rho_*}(q) + \frac{1}{\phi_{\rho_*}(q)}\right)\right)$$
$$- \frac{1}{\pi}\int_0^\pi f_2(k,p)\frac{\sin p}{\phi_{\rho_*}(p) + \frac{1}{\phi_{\rho_*}(p)}}f_2(p,q)\mathrm{d}p = 0, \tag{71}$$

while for $n > 2$ we obtain the linear equation on $f_n$

$$f_n(q_1,\ldots,q_n)\sum_{i=1}^{n}\left(\phi_{\rho_*}(q_i)+\frac{1}{\phi_{\rho_*}(q_i)}\right)\sin q_i - \frac{1}{\pi}\sum_{i=1}^{n}\int_0^\pi \frac{f_2(k,q_i)\sin k}{\phi_{\rho_*}(k)+\frac{1}{\phi_{\rho_*}(k)}}f_n(q_1,\ldots,\underbrace{k}_{i\text{th}},\ldots,q_n)\mathrm{d}k$$
$$= G_{f_2,\ldots,f_{n-1}}(q_1,\ldots,q_n),\tag{72}$$

with some function $G_{f_2,\ldots,f_{n-1}}$ that depends on $f_2,\ldots,f_{n-1}$ and that is symmetric in $q_1,\ldots,q_n$.

As a quadratic equation in $f_2$, (71) has many solutions. To go further, let us express the amplitudes $|\langle\bar{\boldsymbol{k}}|\phi\rangle|^2$ in terms of $f_2$ at next-to-leading order in $\delta\rho$. We consider a density $\rho \neq \rho_*$ and a state $|\bar{\boldsymbol{k}}\rangle_t$ with density $t\rho + (1-t)\rho_*$, and define $b_t$ as in (46), namely the number such that

$$|_t\langle\bar{\boldsymbol{k}}|\phi\rangle|^2 \sim e^{Lb_t}.\tag{73}$$

From (47), one finds

$$\partial_t b_t = 2\int_0^\pi \delta\rho(k)\log\left|\phi_{t\rho+(1-t)\rho_*}(k)\right|\mathrm{d}k,\tag{74}$$

which is, at order $\delta\rho^2$

$$\partial_t b_t = 2\int_0^\pi \delta\rho(k)\log|\phi_{\rho_*}(k)|\mathrm{d}k + 2t\int_0^\pi\int_0^\pi \delta\rho(k)\delta\rho(q)\Re f_2(k,q)\mathrm{d}k\mathrm{d}q.\tag{75}$$

Hence, integrating between 0 and 1, we have $|\langle\bar{\boldsymbol{k}}|\phi\rangle|^2 \sim e^{Lb[\rho]}$ with

$$b[\rho] - b[\rho_*] = 2\int_0^\pi \delta\rho(k)\log|\phi_{\rho_*}(k)|\mathrm{d}k + \int_0^\pi\int_0^\pi \delta\rho(k)\delta\rho(q)\Re f_2(k,q)\mathrm{d}k\mathrm{d}q + \mathcal{O}(\delta\rho^3).\tag{76}$$

Taking into account the entropy factor $e^{LS[\rho]}$, one finds that the states with density $\rho$ contribute to order $e^{Lw[\rho]}$ with

$$w[\rho] - w[\rho_*] = -\frac{1}{2}\int_0^\pi\int_0^\pi \delta\rho(k)\delta\rho(q)\mathcal{H}(k,q)\mathrm{d}k\mathrm{d}q + \mathcal{O}(\delta\rho^3),\tag{77}$$

where

$$\mathcal{H}(p,q) = \frac{\delta(k-q)}{\rho_*(k)(1-2\pi\rho_*(k))} - 2\Re f_2(k,q).\tag{78}$$

We recall from Section 3.4 that the density $\rho_*$ should be a *maximum* of $w[\rho]$, whereas the equation (53) only imposes an extremality condition. This means that for the construction of Section 3.3 to hold, there should be a solution $f_2$ to (71) such that $\mathcal{H}$ is positive definite. To investigate this, we will work with the assumption

$$0 \leq v(k) \leq \frac{1}{4\pi},\tag{79}$$

which is indeed satisfied for zero and finite temperature equilibrium, see Section 4. We find the following criterion

**Property 1.** *Under the assumption* (79), *there is a solution $f_2$ to* (71) *such that* (78) *is positive definite, if and only if*

$$\begin{aligned}&\lambda \geq 0 \text{ and } x \text{ local maximum of } \mathcal{E}(v) \text{ or}\\ &\lambda < 0 \text{ and } x \text{ local minimum of } \mathcal{E}(v).\end{aligned}\tag{80}$$

*Moreover, if it exists, the solution $f_2$ is unique.*

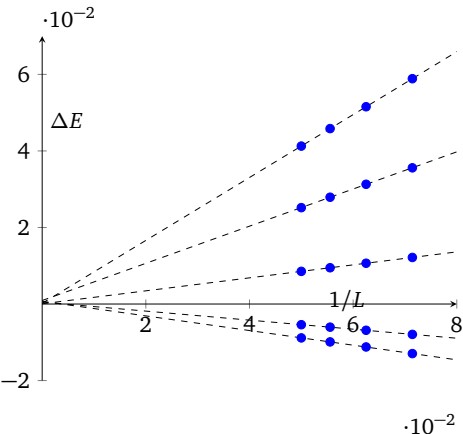

Figure 1: Plot of $\Delta E = (E_{\text{mes}} - E_{\text{th}})/E_{\text{th}}$ as a function of $1/L$, with $E_{\text{mes}}$ the measured ground state energy in size $L$ and $E_{\text{th}}$ the theoretical result for $L \to \infty$. From top to bottom, the parameters used are $(h, \lambda) = (-0.5, -3), (0.5, -1), (2, -0.5), (2, 0.5), (0.5, 1)$. The dashed lines are simple linear fits on the values plotted.

This Property is proven in Appendix B. We note that in the case (79), when $\lambda \geq 0$ a solution to $\partial_x \mathcal{E}(v) = 0$ always satisfies $\partial_x^2 \mathcal{E}(v) < 0$, so the first possibility of Property 1 could be simplified to $\lambda \geq 0$. We also remind the reader that the mean-field equation on the effective magnetic field $x$ derived in Section 3.6 is only that $x$ is an extremum of $\mathcal{E}(v)$. We see here that the maximality of $\rho_*$ refines this condition and imposes a maximality/minimality condition on $x$.

Let us finally briefly comment on physical situations that will involve the full functional $\phi_\rho$. Although the values of $\phi_\rho$ for $\rho \neq \rho_*$ do not play any role in thermodynamic properties, they will appear in broader problems like out-of-equilibrium physics. Indeed, when decomposing a state in terms of the eigenstates of the model, the overlaps are not necessarily dominated by the dominant density $\rho_*$. For example, time-evolving $|0\rangle$ with $H(h, \lambda)$, the overlaps between the initial state and the eigenstates depend only on $|\langle 0|\phi\rangle|^2$, and not on $|\langle \bar{k}|\phi\rangle|^2$ for $\boldsymbol{k}$ with density $\rho_*$.

### 3.9 Numerical checks

In Figure 1 we provide numerical checks of the exact spectrum of (1) in the thermodynamic limit given in Section 3.6. Specifically, we compute numerically the ground state energy density for several parameters and compare it to (62) for $v = 0$ (see below in Section 4.1). We observe excellent agreement, with a relative precision of order $10^{-4}$.

### 3.10 Non-empty set of single momenta $\boldsymbol{s}$

As promised in Section 3.1, let us come back to the case where $\boldsymbol{s}$ the set of single momenta that parametrizes different sectors of $H(h, \lambda)$ is not empty $\boldsymbol{s} \neq \emptyset$. In this case, we define coherent states $|\phi\rangle$ for $\phi(q)$ a function of $q \in K_+^{\boldsymbol{s}}$ as

$$|\phi\rangle = A \sum_{\boldsymbol{k} \subset K_+^{\boldsymbol{s}}} \left( \prod_{k \in \boldsymbol{k}} i\phi(k) \right) |\bar{\boldsymbol{k}} \cup \boldsymbol{s}\rangle. \tag{81}$$

They satisfy as well the factorization property

$$\langle \bar{\boldsymbol{k}} \cup \{q, -q\} \cup \boldsymbol{s} | \phi \rangle = i\phi(q)\langle \bar{\boldsymbol{k}} \cup \boldsymbol{s} | \phi \rangle, \tag{82}$$

for any $q \in K_+^{\boldsymbol{s}} \setminus \boldsymbol{k}$. By changing $K_+$ into $K_+^{\boldsymbol{s}}$, it is straightforward to adapt the derivation of Section 3.2 to show that eigenstates of the TFIM $H(h, 0)$ can be found under this form. Specifically, we have

$$\langle \bar{\boldsymbol{k}} \cup \boldsymbol{s} | H(h, 0) | \phi \rangle = E(\boldsymbol{k})\langle \bar{\boldsymbol{k}} \cup \boldsymbol{s} | \phi \rangle, \tag{83}$$

with

$$E(\boldsymbol{k}) = hL - 2\sum_{s \in \boldsymbol{s}} \cos s - 2\sum_{k \in K_+^{\boldsymbol{s}}} \phi(k)\sin k + 2\sum_{q \in \boldsymbol{k}} \left[ -2h - 2\cos q + \left( \phi(q) - \tfrac{1}{\phi(q)} \right)\sin q \right], \tag{84}$$

and imposing independence from $\boldsymbol{k}$ the same condition (33) follows.

As in the case $\boldsymbol{s} = \emptyset$, eigenstates of $H(h, \lambda)$ can be found by promoting $\phi(q)$ in (82) into a functional $\phi_\rho(q)$ of the density $\rho$ of momenta $\boldsymbol{k}$. We introduce as well $\sigma(k)$ the density of single momenta in $\boldsymbol{s}$, defined for $-\pi < k < \pi$. The candidate energy density $e(\rho)$ of (44) becomes then

$$\begin{aligned}
e(\rho) = {}& h - 2\int_{-\pi}^{\pi} \sigma(k)\cos k\,dk - \frac{1}{\pi}\int_0^\pi (1 - 2\pi(\sigma(k) + \sigma(-k)))\,\phi_\rho(k)\sin k\,dk \\
& + 2\int_0^\pi \rho(k)\left[ -2h - 2\cos k + \left( \phi_\rho(k) - \tfrac{1}{\phi_\rho(k)} \right)\sin k \right]dk \\
& + \lambda\left( 1 - 2\int_{-\pi}^\pi \sigma(k)dk - 4\int_0^\pi \rho(k)dk \right)^2.
\end{aligned} \tag{85}$$

Proceeding as before we obtain the following. We introduce for $-\pi < k < \pi$ the density of excitations $0 \le \nu(k) \le \frac{1}{2\pi}$ counting both the density of the $|k|$'s for which the $-$ sign is chosen in (55), as well as the density of $\boldsymbol{s}$. We find that the energy density $\mathcal{E}(\nu)$ reads

$$\mathcal{E}(\nu) = -\frac{1}{2\pi}\int_{-\pi}^\pi \varepsilon_x(k)(1 - 4\pi\nu(k))\,dk - \frac{(x-h)^2}{4\lambda}, \tag{86}$$

where $x$ satisfies the equation

$$x + \frac{\lambda}{\pi}\int_{-\pi}^\pi \partial_x \varepsilon_x(k)(1 - 4\pi\nu(k))\,dk = h. \tag{87}$$

## 4 Thermodynamics

### 4.1 Phase diagram at zero temperature

#### 4.1.1 Ground state energy

Let us determine the phase diagram of the model at zero temperature, namely the analyticity regions of the ground state energy as a function of $h, \lambda$. With the notations of Section 3.6, the set of excitations $\nu$ corresponding to the ground state has to satisfy the local minimum conditions

$$\begin{cases}
\partial_{\nu(k)}\mathcal{E}(\nu) \ge 0, & \text{if } \nu(k) = 0, \\
\partial_{\nu(k)}\mathcal{E}(\nu) \le 0, & \text{if } \nu(k) = \frac{1}{2\pi}, \\
\partial_{\nu(k)}\mathcal{E}(\nu) = 0, & \text{if } 0 < \nu(k) < \frac{1}{2\pi}.
\end{cases} \tag{88}$$

From (62) and (63), we find

$$\partial_{\nu(k)}\mathcal{E}(\nu) = 4\varepsilon_x(k),\tag{89}$$

which is strictly positive for $0 < k < \pi$. Hence the ground state energy density is obtained with $\nu = 0$ for all parameters $h, \lambda$. So it is

$$F(x) = -\frac{1}{\pi}\int_0^\pi \varepsilon_x(k)\mathrm{d}k - \frac{(x-h)^2}{4\lambda},\tag{90}$$

where $x$ satisfies $F'(x) = 0$. In case of multiple solutions to $F'(x) = 0$, the ground state is given by the lowest $F(x)$. One notes, however, that if there is only one $x$ that satisfies $F'(x) = 0$, it can a priori be the maximum of $F$ and not the minimum.

As a function of $x$, $F(x)$ is non analytical only when $x = \pm 1$. Hence the phase transitions of the model are located either (i) at $(h, \lambda)$ such that $x = \pm 1$, or (ii) when $x$ has a non-analyticity as a function of $h, \lambda$. To go further, we need to study the solutions to $F'(x) = 0$.

### 4.1.2 Case $\lambda \geq 0$

Let us consider first $\lambda \geq 0$. We have

$$\begin{aligned}
F'(x) &= -\frac{1}{\pi}\int_0^\pi \frac{x + \cos k}{\sqrt{x + \cos k)^2 + \sin^2 k}}\mathrm{d}k - \frac{x - h}{2\lambda},\\
F''(x) &= -\frac{1}{\pi}\int_0^\pi \frac{\sin^2 k}{((x + \cos k)^2 + \sin^2 k)^{3/2}}\mathrm{d}k - \frac{1}{2\lambda}.
\end{aligned}\tag{91}$$

We see that $F''(x) < 0$ for all $x$, and $F'(\pm\infty) \to \mp\infty$, hence there is a unique solution to the equation $F'(x) = 0$. It follows that when $\lambda \geq 0$, the only possible phase transitions occur when $x = \pm 1$. Since $F'(\pm 1) = \mp\frac{2}{\pi} - \frac{\pm 1 - h}{2\lambda}$, we conclude that there are two critical lines for $\lambda \geq 0$ given by

$$h \mp \frac{4\lambda}{\pi} = \pm 1.\tag{92}$$

These phase transitions are not associated to discontinuous changes of eigenstates, so they are continuous (second-order) phase transitions. The order parameter is the magnetization $\langle\sigma^z\rangle$ as in the usual TFIM. It is non-zero for $-1 - \frac{4\lambda}{\pi} < h < 1 + \frac{4\lambda}{\pi} \equiv h_c$ and zero for $h$ outside this interval. In the TFIM, the magnetization behaves as $\langle\sigma^z\rangle \sim \delta h^{1/8}$ at the transition at $h = 1 + \delta h$. Here, for $\lambda > 0$, this critical behaviour is marginally modified by the singular behaviour of the effective magnetic field $x$ at the transition. Indeed, around $x = 1 + \delta x$ we have

$$F'(x) = \frac{h}{2\lambda} - \frac{1}{2\lambda} - \frac{2}{\pi} + \frac{1}{\pi}\delta x \log|\delta x| + \mathcal{O}(\delta x).\tag{93}$$

Hence for $h = 1 + \frac{4\lambda}{\pi} + \delta h$, the unique solution to $F'(x) = 0$ behaves as

$$x = 1 - \frac{\pi}{2\lambda}\frac{\delta h}{\log|\delta h|} + o\left(\frac{\delta h}{\log|\delta h|}\right).\tag{94}$$

It follows that we have a behaviour of the magnetization at $\lambda > 0$ that is marginally corrected compared to $\lambda = 0$ as

$$\langle\sigma^z\rangle \sim \left(\frac{h_c - h}{\log(h_c - h)}\right)^{1/8}.\tag{95}$$

We note that this kind of *multiplicative* logarithmic corrections generally arises when marginal operators perturb the field theory describing a critical point [44, 45].

### 4.1.3 Case $\lambda < 0$

Let us now consider the case $\lambda < 0$. It is convenient to introduce

$$\delta F'(x) = F'(x) + \frac{h}{2|\lambda|}, \tag{96}$$

which is independent of $h$. This way, the solutions to $F'(x) = 0$ correspond to intersections of the graph of the function $\delta F'(x)$ with the horizontal lines $\frac{h}{2|\lambda|}$. This naturally suggests to study the phase diagram of the model by fixing $\lambda < 0$ and varying $h$ from $-\infty$ to $+\infty$. To study the solutions to $\delta F'(x) = \frac{h}{2|\lambda|}$ as a function of $h$, we need more information on the behaviour of this function. As shown in Appendix C, $F'''(x)$ is odd and we have

$$\begin{cases} F'''(x) < 0, & \text{for } 0 < x < 1, \\ F'''(x) > 0, & \text{for } x > 1. \end{cases} \tag{97}$$

Hence $F''(x)$ is decreasing for $0 < x < 1$, goes to $-\infty$ at $x = 1$, is increasing for $x > 1$, and goes to $-\frac{1}{2\lambda} > 0$ when $x \to \infty$. Given that $F''(x)$ is even, we conclude that $F''(x)$ changes sign two times from $x = -\infty$ to $+\infty$ if $F''(0) < 0$, and four times if $F''(0) > 0$. It follows that the equation $F'(x) = 0$ has at most three solutions if $F''(0) < 0$, and at most five if $F''(0) > 0$. If one of these solutions $x$ satisfies $F''(x) < 0$, then since $F(x) \to +\infty$ when $x \to \pm\infty$, there has to be another solution $x' < x$ with $F(x') < F(x)$, and so $x$ cannot correspond to the ground state. Hence there are only at most two possible values of $x$ when $F''(0) < 0$, and at most three when $F''(0) > 0$. When $F''(0) < 0$ there is a unique interval of values of $x$ (specifically, where $F''(x) < 0$) that can never correspond to the ground state for any value of $h$, and when $F''(0) > 0$ there are two such intervals.

Now, since $F''(x) \to -\frac{1}{2\lambda} > 0$ when $x \to \infty$, the function $\delta F'(x)$ is strictly increasing for $|x|$ sufficiently large, and is not divergent anywhere on the real line, so for $|h|$ large enough the equation $\delta F'(x) = \frac{h}{2|\lambda|}$ has only one solution $x$ that behaves as $x \to \pm\infty$ when $h \to \pm\infty$. Hence, because of the intervals of values of $x$ that are excluded, there are either one or two discontinuous changes of $x$ as $h$ goes from $-\infty$ to $\infty$ at fixed $\lambda$. These changes precisely correspond to first-order phase transitions. If $F''(0) < 0$ there is only one excluded interval and so there is exactly one phase transition. If $F''(0) > 0$ there are two excluded intervals that do not contain 0. By symmetry, the ground state has a value $x$ between the two excluded intervals if and only if the ground state is obtained for $x = 0$ at $h = 0$. Hence, summarizing, there is exactly one phase transition from $h = -\infty$ to $h = \infty$ if $x = 0$ is not the minimum of $F(x)$ at $h = 0$, and this phase transition occurs at $h = 0$. There are exactly two phase transitions from $h = -\infty$ to $h = \infty$ if $x = 0$ is the minimum of $F(x)$ at $h = 0$, and they occur at some values of magnetic field $\pm h_0(\lambda)$. Let us investigate quantitatively these conditions as a function of $\lambda$. We have

$$F''(0) = -\frac{1}{2} - \frac{1}{2\lambda}, \tag{98}$$

so at least for $\lambda < -1$ we have $F''(0) < 0$, implying only one phase transition. For $0 < \lambda < -1$ one has $F''(0) > 0$, so that the criterion to distinguish between one or two phase transitions is whether $F(0) = -1$ is the minimum of the function $F(x)$ at $h = 0$. This condition means

$$\forall x, \quad -\frac{1}{\pi} \int_0^\pi \sqrt{1 + x^2 + 2x\cos k}\, dk - \frac{x^2}{4\lambda} \geq -1, \tag{99}$$

which is equivalent to

$$\forall x, \quad \lambda \geq \frac{1}{4} \frac{x^2}{1 - \frac{1}{\pi}\int_0^\pi \sqrt{1 + x^2 + 2x\cos k}\, dk}. \tag{100}$$

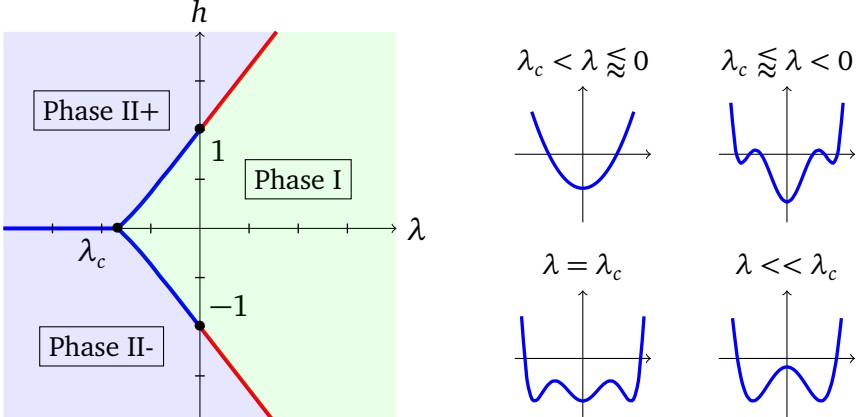

Figure 2: Left: Phase diagram at zero temperature of the Hamiltonian (1) in the $(\lambda, h)$ plane. Blue thick lines indicate first-order phase transitions and red thick lines second-order phase transitions. The value of $\lambda_c$ is quoted in (102). Right: Sketches of $F(x)$ as a function of $x$ for $h = 0$ and different values of $\lambda$. The ground state value is given by the minimum of the function plotted.

Hence, denoting

$$\lambda_c = -\frac{1}{4} \min_{x \in \mathbb{R}} \frac{x^2}{\frac{1}{\pi} \int_0^\pi \sqrt{1 + x^2 + 2x \cos k}\, dk - 1}, \tag{101}$$

there is only one phase transition in $h$ for $\lambda < \lambda_c$, occurring at $h = 0$, and there are two phase transitions in $h$ for $\lambda > \lambda_c$. Numerically, this critical value is

$$\lambda_c = -0.836584\ldots \tag{102}$$

Finally, we remark that when $\lambda < 0$ there cannot be any phase transitions coming from the non-analyticity of $F(x)$ at $x = \pm 1$. Indeed, we have $F''(\pm 1) \to -\infty$, so $x = \pm 1$ cannot be a minimum, so $x = \pm 1$ is never the effective magnetic field corresponding to the ground state.

From these analytic considerations, we obtain the following phase diagram in Figure 2. The different phases mentioned in Figures 2 and 4 are

- Phase I: ordered phase for $\sigma^z$: $\langle \sigma^z \rangle \neq 0$. Paramagnetic phase for $\sigma^x$: $\langle \sigma^x \rangle \neq 0$ increasing odd function of $h$.

- Phase II±: disordered phase for $\sigma^z$: $\langle \sigma^z \rangle = 0$. Paramagnetic phase for $\sigma^x$: $\langle \sigma^x \rangle \neq 0$ increasing odd function of $h$.

- Phase II: disordered phase for $\sigma^z$: $\langle \sigma^z \rangle = 0$. Ordered phase for $\sigma^x$: $\langle \sigma^x \rangle \neq 0$.

- Phase III: disordered phase for $\sigma^z$: $\langle \sigma^z \rangle = 0$, and for $\sigma^x$: $\langle \sigma^x \rangle = 0$.

Moreover, we provide in Figure 3 a numerical comparison of the phase transition for $\langle \sigma_j^x \rangle$ as a function of $h$ at zero temperature.

## 4.2 Phase diagram at finite temperature

### 4.2.1 The free energy at finite temperature

Let us now consider finite temperature equilibrium. It is known that in presence of long-range interactions, the canonical and microcanonical ensembles are not necessarily equivalent [46].

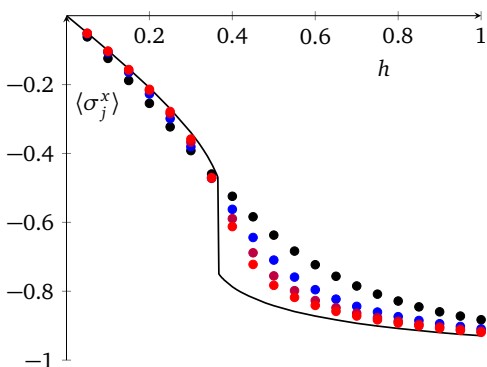

Figure 3: Plot of $\langle \sigma_j^x \rangle$ evaluated in the ground state of the model across a first order phase transition, as a function of $h$ for $\lambda = -0.5$, for different sizes $L = 6, 10, 14, 18$ (black, blue, purple, red). The thick line is the theoretical result in the thermodynamic limit, and displays a first-order phase transition.

We will restrict our study of the finite temperature phase diagram to the canonical ensemble only. In this case, the free energy at finite inverse temperature $\beta$ is defined by

$$\mathfrak{f}(\beta) = \lim_{L\to\infty} -\frac{1}{\beta L} \log \operatorname{tr} e^{-\beta H(h,\lambda)}. \tag{103}$$

In the partition function $\operatorname{tr} e^{-\beta H(h,\lambda)}$, the terms with density of excitations $\nu$ contribute as $e^{-\beta L \mathcal{E}(\nu)}$ with $\mathcal{E}(\nu)$ the energy density given in (62), and there are asymptotically $e^{LS(\nu)}$ of them, with the entropy

$$S(\nu) = -\int_0^\pi \left[ \nu(k)\log(2\pi\nu(k)) + \left(\tfrac{1}{2\pi} - \nu(k)\right)\log(1 - 2\pi\nu(k)) \right] dk. \tag{104}$$

Hence the free energy is

$$\mathfrak{f}(\beta) = \min_\nu \left[ \mathcal{E}(\nu) - \frac{1}{\beta} S(\nu) \right], \tag{105}$$

where the minimum is taken over the set of functions $0 \le \nu(k) \le \frac{1}{2\pi}$. The extremality condition for the minimal $\nu$ is

$$\nu(k) = \frac{1}{2\pi} \frac{1}{1 + e^{\beta \partial_{\nu(k)} \mathcal{E}(\nu)}}, \tag{106}$$

with $\partial_{\nu(k)} \mathcal{E}(\nu)$ given in (89). Hence, writing $\mathcal{E}(\nu) - \frac{1}{\beta} S(\nu)$ in terms of this $\nu(k)$, one finds that the free energy is

$$F_\beta(x) = -\frac{1}{2\pi\beta} \int_0^\pi \log[2\cosh(2\beta\varepsilon_x(k))] dk - \frac{(x-h)^2}{4\lambda}, \tag{107}$$

with $x$ satisfying (63), which is exactly $F_\beta'(x) = 0$. In case of multiple solutions $x$, the free energy is the one that minimizes $F_\beta(x)$ among these solutions. One notes, however, that as in the zero-temperature case, $x$ can be a local maximum of $F_\beta(x)$ if there is only one $x$ that satisfies $F_\beta'(x) = 0$.

### 4.2.2 Case $\lambda \geq 0$

We start with the case $\lambda \geq 0$. Looking for solutions to $F'_\beta(x) = 0$, we compute

$$
\begin{aligned}
F''_\beta(x) = &-\frac{1}{2\lambda} - \frac{1}{\pi} \int_0^\pi \partial_x^2 \varepsilon_x(k) \tanh(2\beta \varepsilon_x(k)) \mathrm{d}k \\
&- \frac{2\beta}{\pi} \int_0^\pi (\partial_x \varepsilon_x(k))^2 \left(1 - \tanh^2(2\beta \varepsilon_x(k))\right) \mathrm{d}k \,.
\end{aligned}
\tag{108}
$$

Since we always have $\varepsilon_x(k), \partial_x^2 \varepsilon_x(k) \geq 0$, we see that for $\lambda \geq 0$ we have $F''_\beta(x) < 0$ for all $x$. Since $F'_\beta(\pm\infty) \to \mp\infty$, we conclude that there is a unique solution to $F'_\beta(x)=0$ and that solution is a smooth function of $h, \lambda, \beta$. Moreover, for $\beta < \infty$ the free energy $F_\beta(x)$ is a smooth function of $x$. Hence there is no phase transition at finite temperature for $\lambda \geq 0$, and the system is in the high-temperature phase as soon as $\beta < \infty$.

### 4.2.3 Case $\lambda < 0$ and $\beta$ small

We now consider $\lambda < 0$ and the high temperature phase, namely $\beta$ close to 0 in a sense that we will precise. Using the expression for $\varepsilon_x(k)$, we write

$$
\begin{aligned}
F''_\beta(x) = &-\frac{1}{2\lambda} - \frac{2\beta}{\pi} \int_0^\pi \left(1 - (\partial_x \varepsilon_x(k))^2\right) \frac{\tanh(2\beta \varepsilon_x(k))}{2\beta \varepsilon_x(k)} \mathrm{d}k \\
&- \frac{2\beta}{\pi} \int_0^\pi (\partial_x \varepsilon_x(k))^2 \left(1 - \tanh^2(2\beta \varepsilon_x(k))\right) \mathrm{d}k \,.
\end{aligned}
\tag{109}
$$

Using then $|\frac{\tanh x}{x}| \leq 1$ and $|\partial_x \varepsilon_x(k)| \leq 1$, we find that for $\lambda < 0$ and

$$
\beta < \frac{1}{8|\lambda|} \,,
\tag{110}
$$

we have $F''_\beta(x) > 0$ for all $x$, so only one solution to $F'_\beta(x) = 0$. Hence at least in the region (110), the system is in the high-temperature phase.

This also implies that at fixed $\beta < \infty$, there is always a band $\lambda_0(\beta) < \lambda < 0$ with $\lambda_0(\beta) < -\frac{1}{8\beta}$ in which the system is in the high-temperature phase.

### 4.2.4 Case $\lambda < 0$ and $\beta$ large

We now fix $\lambda < 0$ and $h$ and consider the low temperature phase, namely $\beta \gg 1$ in a sense that we will precise. Comparing the finite temperature $F'_\beta(x)$ and the zero temperature $F'(x)$ we have

$$
F'_\beta(x) - F'(x) = \frac{1}{\pi} \int_0^\pi \partial_x \varepsilon_x(k)(1 - \tanh(2\beta \varepsilon_x(k))) \mathrm{d}k \,.
\tag{111}
$$

Hence

$$
|F'_\beta(x) - F'(x)| \leq e^{-4\beta \min(|1-x|, |1+x|)} \,,
\tag{112}
$$

and it follows that for any fixed $\delta > 0$, $F'_\beta(x)$ converges uniformly to $F'(x)$ in the region $||x| - 1| > \delta$ when $\beta \to \infty$. Now, we know that $F'(\pm 1) \to -\infty$ and that for $\lambda < 0$ the value of $x$ corresponding to the ground is a minimum of $F(x)$ (see the end of Section 4.1.3). Hence at fixed $\lambda < 0$, there is a $\delta > 0$ such that for all $h$, the value of $x$ at zero temperature satisfies $||x| - 1| > \delta$. It follows that solutions to $F'_\beta(x) = 0$ have a smooth dependence in $\beta$ for $\beta$ in a neighbourood of $\infty$.

Let us now assume that $h, \lambda$ are chosen away from the phase transition lines. In this case there is a unique global minimum of $F(x)$, and the other possible local minima take a value $\delta' > 0$ larger. Hence for $\beta$ large enough the minimum of the function $F_\beta(x)$ will vary smoothly in $\beta$. We conclude thus that for $\lambda < 0$ and $h, \lambda$ away from the phase transition lines at zero temperature, there is no phase transition in $x$ for $\beta_0(h, \lambda) < \beta \leq \infty$ with some large enough $\beta_0(h, \lambda) < \infty$.

### 4.2.5 Existence of a finite-temperature phase transition for $\lambda < 0$ and $h = 0$

Let us fix $\lambda < 0$ and consider the case $h = 0$. From Section 4.2.3 we know that for $\beta$ small enough the only solution to $F'_\beta(x) = 0$ is $x = 0$. Now, we have

$$F''_\beta(0) = -\frac{1}{2\lambda} - \frac{\tanh(2\beta)}{2} - \beta(1 - \tanh^2(2\beta)). \tag{113}$$

So $F''_\beta(0) \geq 0$ for all $\beta$ is equivalent to

$$\lambda \geq \lambda'_c, \tag{114}$$

with

$$\lambda'_c = -\min_{\beta > 0} \frac{1}{\tanh(2\beta) + 2\beta(1 - \tanh^2(2\beta))}. \tag{115}$$

The value $\beta'_c$ for which this minimum is attained is the unique positive solution to

$$2\beta'_c \tanh(2\beta'_c) = 1. \tag{116}$$

Numerically, this is

$$\lambda'_c = -0.833557\ldots, \qquad \beta'_c = 0.599839\ldots \tag{117}$$

Hence for $\lambda < \lambda'_c$ there is necessarily a range of temperatures for which $F''_\beta(0) < 0$, hence for which $x = 0$ is not a minimum of $F_\beta(x)$. Hence at least for $\lambda < \lambda'_c$ there is a phase transition at finite temperature.

### 4.2.6 More details: numerical study

To say more about the phase diagram, we need to carry out a precise analytical study of $F_\beta(x)$, similar to that performed in the zero-temperature case. Unfortunately, the finite temperature case is significantly more involved and we were not able to prove the following result generalizing (97). What we observe numerically is the existence of $\beta''_c > 0$ that satisfies $F'''_{\beta''_c}(0) = 0$ and that is such that

for $\beta < \beta''_c$, $\quad \forall x > 0$, $\quad F'''_\beta(x) > 0$,

for $\beta > \beta''_c$, there exists a unique $0 < x_0 < 1$ such that $\begin{cases} F'''_\beta(x) < 0, & \text{for } 0 < x < x_0, \\ F'''_\beta(x) > 0, & \text{for } x > x_0. \end{cases}$

$$\tag{118}$$

We will assume (118) satisfied in the following. Let us investigate the consequences of this property.

**Second-order finite temperature phase transition line**

We know that at fixed $\lambda < \lambda'_c$ there is a phase transition at finite temperature, since $F''_\beta(0)$ is positive for $\beta = 0$ and negative for $\beta$ large enough. However, the transition does not necessarily occur at $x = 0$, as there could be another solution $x > 0$ to $F'_\beta(x) = 0$ appearing as we increase $\beta$ before $F''_\beta(0)$ becomes negative. Because of (118), a necessary and sufficient

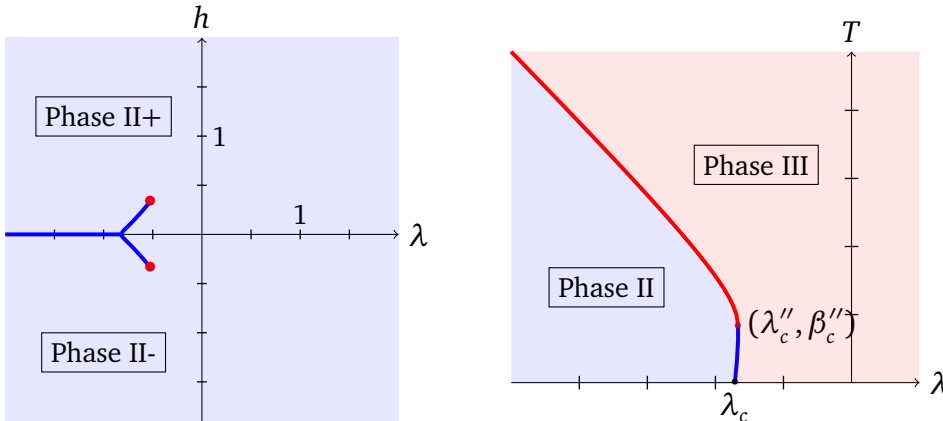

Figure 4: Left: phase diagram at temperature $T = 0.33$ with the same conventions as the left panel of Figure 2. The red dots indicate critical points with second-order phase transition, occuring at $\approx (-0.525, \pm 0.335)$. Right: sketch of the finite temperature phase diagram of $H(h, \lambda)$ for $h = 0$, in terms of $\lambda$ and temperature $T = 1/\beta$. The numerical values are quoted in (120) and (102). The blue thick line indicates a first-order phase transition and the red thick line a second-order phase transition whose expression is (121). The curve of the blue line has been slightly exaggerated on the plot so that the inverse melting region is visible.

condition for this latter fact not to happen is that $F'''_{\beta_0(\lambda)}(0) > 0$ at the value of $\beta = \beta_0(\lambda)$ for which $F''_{\beta_0(\lambda)}(0) = 0$. Hence if $\beta_0(\lambda) < \beta''_c$ then there is a second-order phase transition at $\beta_0(\lambda)$. Otherwise the transition is first order. We define then $\lambda''_c$ by $F''_{\beta''_c}(0) = 0$, i.e.,

$$\lambda''_c = -\frac{1}{\tanh(2\beta''_c) + 2\beta''_c(1 - \tanh^2(2\beta''_c))}. \tag{119}$$

Numerically, we find

$$\lambda''_c = -0.834428\ldots, \quad \beta''_c = 0.571496\ldots \tag{120}$$

For $\lambda < \lambda''_c$ there is always a second-order phase transition at finite temperature $\beta$ satisfying $F''_\beta(0) = 0$, namely given by the curve

$$\lambda = -\frac{1}{\tanh(2\beta) + 2\beta(1 - \tanh^2(2\beta))}, \quad \lambda < \lambda''_c. \tag{121}$$

**First-order phase transition line at finite temperature**

We see that $\lambda''_c < \lambda'_c$. Hence there is a region that includes at least $\lambda''_c < \lambda < \lambda'_c$ for which there is a first-order phase transition at finite temperature as we increase $\beta$ occurring before (and so, hiding) the second-order phase transition.

**Inverse melting/freezing region**

We finally note that $\lambda_c < \lambda'_c$. At zero temperature for $\lambda > \lambda_c$ and $h = 0$ the system is ordered for $\sigma^z$ and disordered for $\sigma^x$. We know from Section 4.2.4 that at low temperature the system will remain disordered for $\sigma^x$ (and becomes immediately disordered for $\sigma^z$). At high temperature the system is also disordered for $\sigma^x$. However, we also know that for $\lambda < \lambda'_c$ there is necessarily a phase transition for $\sigma^x$ as we lower the temperature from the infinite

temperature region, entering thus an ordered phase for $\sigma^x$. Hence at least for $\lambda_c < \lambda < \lambda'_c$ there is inverse melting/freezing [15], in the sense that increasing/decreasing the temperature drives the system into an ordered/disordered phase. We note that the region of parameter space $[\lambda_c, \lambda'_c]$ is very tiny, but the inverse melting/freezing regions of other models are also typically tiny [16, 18]. Gathering these different findings, the phase diagram of the model at finite temperature is plotted in Figure 4.

## 5 Summary and discussion

In this paper, we showed that the transverse field Ising model with additional all-to-all interactions can be exactly solved with MFT in the thermodynamic limit. This is a surprising fact as the model contains short-range interactions, which usually spoils the exactness of MFT. We obtained an expression for the energy density of any state in the thermodynamic limit, as well as expectation value of local operators within any state. We then studied the phase diagram of the model at both zero and finite temperature. These results are interesting for the following reasons.

Firstly, this work provides a new solvable model that is in a sense intermediate between a free model and an interacting model. In this optics it can be used as a testbed for studying features of many-body quantum physics. This is particularly relevant since it displays properties that are absent from the TFIM while remaining solvable, most notably a phase transition at finite temperature. Such transitions are usually absent from 1D models from Peierls's argument, and 2D quantum models which can display them present huge obstacles. Hence this model could open valuable analytical studies. Among other facts, the model also displays critical behaviour with a marginally relevant perturbation, as well as a region with inverse melting/freezing. An interesting open question is whether the model displays thermalization like a generic many-body quantum model, or if it behaves like an integrable model.

Secondly, we introduced a novel method to solve the model in the thermodynamic limit. It consists in looking for an eigenstate of the model under the form of an eigenstate of a free model whose parameters are modulated by the density of momenta in a basis where the Hamiltonian only creates a finite number of excitations. The fact that the method only works in the thermodynamic limit is appealing, since it leaves hope for more general applicability beyond this model. However, importantly, the present method also relies on the exact solvability of the MFT Hamiltonian. For these reasons, we plan to investigate in future works to what extent the method can be applied to more general models.

## Acknowledgments

We are grateful to M. Levin for useful discussions and comments. We thank F.H.L. Essler for comments on the draft. This work was supported by the Kadanoff Center for Theoretical Physics at University of Chicago, and by the Simons Collaboration on Ultra-Quantum Matter.

## A Proof of (43)

**Property 2.** *We consider $\phi_{\boldsymbol{k}}(q)$ a functional of $\boldsymbol{k}$ that is assumed to depend only on $\rho$ the density of momenta of $\boldsymbol{k}$ in the thermodynamic limit, then denoted $\phi_{\rho}(q)$, and with a smooth dependence*

in $\rho$. For $k_1, \ldots, k_{n_L} \in K$ distinct with $n_L = O(L)$, we define the quantity

$$Z(k_1, \ldots, k_{n_L}) = \prod_{j=1}^{n_L} \phi_{\{k_1, \ldots, k_{j-1}\}}(k_j). \tag{A.1}$$

We have for all $k, q$

$$\frac{\partial_{\rho(q)} \phi_\rho(k)}{\phi_\rho(k)} = \frac{\partial_{\rho(k)} \phi_\rho(q)}{\phi_\rho(q)}, \tag{A.2}$$

if and only if we have

$$\frac{Z(k_1, \ldots, k_p, \ldots, k_q, \ldots, k_{n_L})}{Z(k_1, \ldots, k_q, \ldots, k_p, \ldots, k_{n_L})} = 1 + O(L^{-1}), \tag{A.3}$$

for all sequences $k_1, \ldots, k_{n_L}$ and any $p < q$.

*Proof.* We will denote $\rho_j$ the state with momenta $k_1, \ldots, k_{j-1}$, and $Z/Z'$ the ratio studied. We have

$$
\begin{aligned}
\frac{Z}{Z'} &= \frac{\phi_{\rho_p}(k_p)}{\phi_{\rho_p}(k_q)} \left( \prod_{j=p+1}^{q-1} \frac{\phi_{\rho_j}(k_j)}{\phi_{\rho_j \setminus \{k_p\} \cup \{k_q\}}(k_j)} \right) \frac{\phi_{\rho_q}(k_q)}{\phi_{\rho_q \setminus \{k_p\} \cup \{k_q\}}(k_p)} \\
&= \frac{\phi_{\rho_p}(k_p)}{\phi_{\rho_p}(k_q)} \left( \prod_{j=p+1}^{q-1} \frac{\phi_{\rho_j}(k_j)}{\phi_{\rho_j}(k_j) - \frac{\partial_{\rho(k_p)} \phi_{\rho_j}(k_j)}{L} + \frac{\partial_{\rho(k_q)} \phi_{\rho_j}(k_j)}{L}} \right) \frac{\phi_{\rho_q}(k_q)}{\phi_{\rho_q}(k_p)} + O(L^{-1}) \\
&= \frac{\phi_{\rho_p}(k_p) \phi_{\rho_q}(k_q)}{\phi_{\rho_p}(k_q) \phi_{\rho_q}(k_p)} \exp \left[ \frac{1}{L} \sum_{j=p+1}^{q-1} \partial_{\rho(k_p)} \log \phi_{\rho_j}(k_j) - \partial_{\rho(k_q)} \log \phi_{\rho_j}(k_j) \right] + O(L^{-1}).
\end{aligned} \tag{A.4}
$$

If (A.2) is satisfied, then

$$
\begin{aligned}
\frac{Z}{Z'} &= \frac{\phi_{\rho_p}(k_p) \phi_{\rho_q}(k_q)}{\phi_{\rho_p}(k_q) \phi_{\rho_q}(k_p)} \exp \left[ \frac{1}{L} \sum_{j=p+1}^{q-1} \partial_{\rho(k_j)} \log \phi_{\rho_j}(k_p) - \partial_{\rho(k_j)} \log \phi_{\rho_j}(k_q) \right] + O(L^{-1}) \\
&= \frac{\phi_{\rho_p}(k_p) \phi_{\rho_q}(k_q)}{\phi_{\rho_p}(k_q) \phi_{\rho_q}(k_p)} \exp \left[ \log \frac{\phi_{\rho_q}(k_p)}{\phi_{\rho_p}(k_p)} - \log \frac{\phi_{\rho_q}(k_q)}{\phi_{\rho_p}(k_q)} \right] + O(L^{-1}) \\
&= 1 + O(L^{-1}),
\end{aligned} \tag{A.5}
$$

because $\log \phi_{\rho_{j+1}}(k_p) = \log \phi_{\rho_j}(k_p) + \frac{1}{L} \partial_{\rho(k_j)} \log \phi_{\rho_j}(k_p)$. This shows the direction $\implies$ of the equivalence.

If now (A.2) is not satisfied at some point, then there is $\epsilon_0 > 0$ such that for any $0 < \epsilon < \epsilon_0$ there exist intervals $I, J$ such that for $k \in I, q \in J$

$$\frac{\partial_{\rho(q)} \phi_\rho(k)}{\phi_\rho(k)} - \frac{\partial_{\rho(k)} \phi_\rho(q)}{\phi_\rho(q)} > \epsilon. \tag{A.6}$$

Taking $I$ small enough, we can always assume besides that for $k, q \in I$

$$\left| \frac{\partial_{\rho(q)} \phi_\rho(k)}{\phi_\rho(k)} - \frac{\partial_{\rho(k)} \phi_\rho(q)}{\phi_\rho(q)} \right| < \epsilon^2, \tag{A.7}$$

because (A.2) is satisfied if $k = q$. Then, considering a sequence $k_1, \ldots, k_{n_L}$ such that $k_p \in J, k_q \in I$ and $k_j \in I$ for all $p < j < q$, together with $q - p = cL$ with $c > 0$, we have

$$\frac{Z}{Z'} > \frac{\phi_{\rho_p}(k_p)\phi_{\rho_q}(k_q)}{\phi_{\rho_p}(k_q)\phi_{\rho_q}(k_p)} \exp\left[\epsilon c + O(\epsilon^2) + \frac{1}{L}\sum_{j=p+1}^{q-1} \partial_{\rho(k_j)}\log\phi_{\rho_j}(k_p) - \partial_{\rho(k_j)}\log\phi_{\rho_j}(k_q)\right]$$

$$+ O(L^{-1})$$
$$> e^{\epsilon c + O(\epsilon^2)} + O(L^{-1}),$$

(A.8)

which cannot be $1 + O(L^{-1})$ for $\epsilon$ small enough. This shows the other direction of the equivalence. □

## B  Proof of Property 1

Let us first do the change of variable

$$f_2(k,q) = -g(k,q)\frac{\phi_{\rho_*}(q) + \frac{1}{\phi_{\rho_*}(q)}}{\sin q} + \pi(\phi_{\rho_*}(q) + \frac{1}{\phi_{\rho_*}(q)})^2\delta(k-q).$$

(B.1)

The equation (71) becomes

$$\int_0^\pi g(k,p)g(p,q)\mathrm{d}p = \pi^2\left(\phi_{\rho_*}(q) + \frac{1}{\phi_{\rho_*}(q)}\right)^2\sin^2 q\,\delta(k-q) + 16\pi\lambda\frac{\sin q}{\phi_{\rho_*}(q) + \frac{1}{\phi_{\rho_*}(q)}}.$$

(B.2)

Hence $g(k,q)$ is a square root of the matrix on the right-hand side. As a diagonal matrix plus a rank one matrix, the right-hand side is diagonalizable. So for each eigenvalue there is a $\pm$ sign to choose when considering a solution $g$. There are thus several solutions to the equation (71). In terms of $g$, $\mathcal{H}$ is then

$$\mathcal{H}(k,q) = 2\Re g(k,q)\frac{\phi_{\rho_*}(q) + \frac{1}{\phi_{\rho_*}(q)}}{\sin q}.$$

(B.3)

We now introduce the following matrix for $0 \leq t \leq 1$

$$G_t(k,q) = \pi^2\left(\phi_{\rho_*}(q) + \frac{1}{\phi_{\rho_*}(q)}\right)^2\sin^2 q\,\delta(k-q) + 16\pi\lambda t\frac{\sin q}{\phi_{\rho_*}(q) + \frac{1}{\phi_{\rho_*}(q)}}.$$

(B.4)

The matrix on the right-hand side of (B.2 ) is $G_1$, and $G_t$ has a smooth dependence on $t$. Let us show that $G_t(k,q)$ is positive definite for all $0 \leq t \leq 1$ if the condition of Property 1 is satisfied, under the assumption (79), and that $G_1$ is not positive definite if the condition of Property 1 is not satisfied.
Using that

$$\sin q\left(\phi_{\rho_*}(q) + \frac{1}{\phi_{\rho_*}(q)}\right) = 2\mathfrak{s}(q)\varepsilon_x(q),$$

(B.5)

with $\mathfrak{s}(q)$ the $\pm$ sign appearing in (55), we have

$$G_t(p,q) = 4\pi^2\varepsilon_x(q)^2\delta(k-q) + 8\pi\lambda t\frac{\mathfrak{s}(q)\sin^2 q}{\varepsilon_x(q)}.$$

(B.6)

Using the matrix-determinant lemma holding for an invertible matrix $A$ and column vectors $u,v$

$$\det(A + uv^t) = (1 + v^t A^{-1}u)\det A,$$

(B.7)

we find by writing the characteristic polynomial of (B.6 ) that a negative eigenvalue $-z$ of $G_t$ with $z > 0$ must satisfy

$$1 + 8\pi\lambda t \int_0^\pi \mathfrak{s}(k) \frac{\sin^2 k}{\varepsilon_x(k)} \cdot \frac{1}{4\pi^2\varepsilon_x(k)^2 + z} \mathrm{d}k = 0. \tag{B.8}$$

Hence in terms of $v(k)$ we have

$$1 + 8\pi\lambda t \int_0^\pi \frac{\sin^2 k}{\varepsilon_x(k)} \frac{1 - 4\pi v(k)}{4\pi^2\varepsilon_x(k) + z} \mathrm{d}k = 0. \tag{B.9}$$

If $\lambda \geq 0$, using the assumption (79), this equation cannot hold for $z \geq 0$ since the integrand is strictly positive. Hence the matrix on the right-hand side of (B.2 ) has only positive eigenvalues if $\lambda \geq 0$. Let us now consider $\lambda < 0$. Using $z > 0$, $0 \leq t \leq 1$ and the assumption (79), we have

$$\left| 8\pi t\lambda \int_0^\pi \frac{\sin^2 k}{\varepsilon_x(k)} \frac{1 - 4\pi v(k)}{4\pi^2\varepsilon_x(k) + z} \mathrm{d}k \right| < \frac{2|\lambda|}{\pi} \int_0^\pi \frac{\sin^2 k}{\varepsilon_x(k)^3}(1 - 4\pi v(k)) \mathrm{d}k. \tag{B.10}$$

We remark now that

$$\frac{\sin^2 k}{\varepsilon_x(k)^3} = \partial_x^2 \varepsilon_x(k). \tag{B.11}$$

But the minimality condition (1) for $\lambda < 0$ written in terms of $\mathcal{E}(v)$ read

$$\frac{2|\lambda|}{\pi} \int_0^\pi \partial_x^2 \varepsilon_x(k)(1 - 4\pi v(k)) \mathrm{d}k \leq 1. \tag{B.12}$$

Hence we have

$$\left| 8\pi t\lambda \int_0^\pi \frac{\sin^2 k}{\varepsilon_x(k)} \frac{1 - 4\pi v(k)}{4\pi^2\varepsilon_x(k) + z} \mathrm{d}k \right| < 1, \tag{B.13}$$

and so (B.8 ) cannot be. Hence $G_t$ has only strictly positive eigenvalues for any $\lambda$.

Let us now assume that the condition of Property 1 is not satisfied, which implies $\lambda < 0$, and set $t = 1$. At $z = 0$ the left-hand side of (B.9 ) is negative, whereas for $z \to \infty$ it is positive. By continuity there has to be a solution $z > 0$ to (B.9 ), and so $G_1$ has a negative eigenvalue.

We now define $g_t(k, q)$ by

$$\int_0^\pi g_t(k, p) g_t(p, q) \mathrm{d}p = G_t(k, q). \tag{B.14}$$

For $t = 0$, we have

$$g_0(k, q) = s(q)\pi(\phi_{\rho_*}(q) + \frac{1}{\phi_{\rho_*}(q)})\sin q\,\delta(k - q), \tag{B.15}$$

with $s(q) \in \{1, -1\}$ a $q$-dependent sign. Assuming the condition of Property 1 satisfied, $G_t(p, q)$ is always positive definite for $0 \leq t \leq 1$. Hence each of these solutions produces a family of solutions $g_t(k, q)$ that are continuous in $t$, and whose eigenvalues never vanish. So for each solution, the $\mathcal{H}_t$ defined in terms of $g_t$ is smooth in $t$ and its determinant never vanishes. Hence a solution $g_t$ gives a positive definite $\mathcal{H}_t$ at $t = 1$ if and only if it is gives a positive definite $\mathcal{H}_0$ at $t = 0$. And at $t = 0$ only the solution $g_0(k, q) = \pi(\phi_{\rho_*}(q) + \frac{1}{\phi_{\rho_*}(q)})\sin q\,\delta(k - q)$ gives a positive definite $\mathcal{H}_0$. Hence, at $t = 1$ there is a unique solution $g$ such that $\mathcal{H}_1 = \mathcal{H}$ is positive definite.

Assuming now the condition of Property 1 not satisfied, $g$ has to have at least one purely imaginary eigenvalue. But since $G_1$ is a real rank one perturbation of a real diagonal matrix, its eigenvectors are real, and so those of $g$ are real too. Hence $\Re g$ has a zero eigenvalue. It follows that $\mathcal{H}$ has also a zero eigenvalue and it is not positive definite.

## C   Proof of (97)

We have

$$F'''(x) = \frac{3}{\pi} \int_0^\pi \frac{(x + \cos k)\sin^2 k}{((x + \cos k)^2 + \sin^2 k)^{5/2}} dk. \tag{C.1}$$

The change of variable $k \to \pi - k$ shows that it is an odd function of $x$, so that we focus on $x > 0$. For $x > 1$, since $x + \cos k > 0$ for all $k$, we immediately have $F'''(x) > 0$. The case $0 < x < 1$ is more involved. Let us first show that for $0 < x < 1$ we have

$$\int_0^\pi \frac{(x + \cos k)\sin^2 k}{((x + \cos k)^2 + \sin^2 k)^2} dk = 0. \tag{C.2}$$

To that end, we write

$$\int_0^\pi \frac{(x + \cos k)\sin^2 k}{((x + \cos k)^2 + \sin^2 k)^2} dk = -\frac{1}{2}\partial_x \int_0^\pi \frac{\sin^2 k}{(x + \cos k)^2 + \sin^2 k} dk$$

$$= -\frac{1}{4}\partial_x \left[ (1 + x^2) \int_0^{2\pi} \frac{\sin^2 k}{(x + e^{ik})(x + e^{-ik})(x - e^{ik})(x - e^{-ik})} dk \right] \tag{C.3}$$

$$= -\frac{i}{16}\partial_x \left[ (1 + x^2) \oint \frac{(z^2 - 1)^2}{z(x + z)(zx + 1)(x - z)(zx - 1)} dz \right],$$

where the last integral is over the unit circle. The residue theorem gives for $0 < x < 1$

$$\oint \frac{(z^2 - 1)^2}{z(x + z)(zx + 1)(x - z)(zx - 1)} dz = -\frac{4i\pi}{1 + x^2}. \tag{C.4}$$

Hence we obtain (C.2) indeed. Now, we write

$$F'''(x) = \frac{3}{\pi} \int_0^\pi \frac{(x + \cos k)\sin^2 k}{(1 + x^2 + 2x\cos k)^2} \frac{1}{\sqrt{1 + x^2 + 2x\cos k}} dk, \tag{C.5}$$

and notice that the integrand is strictly positive for $k < \arccos(-x) \equiv q$ and strictly negative for $k > q$, while the factor $\frac{1}{\sqrt{1 + x^2 + 2x\cos k}}$ is a strictly increasing function of $k$. This factor takes the value $\frac{1}{\sqrt{1 - x^2}}$ at $k = q$. Hence

$$F'''(x) < \frac{3}{\pi} \frac{1}{\sqrt{1 - x^2}} \int_0^q \frac{|x + \cos k|\sin^2 k}{(1 + x^2 + 2x\cos k)^2} dk - \frac{3}{\pi} \frac{1}{\sqrt{1 - x^2}} \int_q^\pi \frac{|x + \cos k|\sin^2 k}{(1 + x^2 + 2x\cos k)^2} dk, \tag{C.6}$$

which is, using (C.2)

$$F'''(x) < 0. \tag{C.7}$$

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
