# Peer review of "Exact mean-field solution of a spin chain with short-range and long-range interactions"

_SciPost Physics, doi:SciPost Phys. 14, 133 (2023)_

## Round 1 · Referee Report · Anonymous (Referee 1) · 2022-10-13

Strengths

The Hamiltonian (1) is solved by a mean field in the sense that local observables are obtained from a generalized paired state ansatz which, in the thermodynamic limit, makes the problem exactly equivalent to a mean-field Hamiltonian. This is a very interesting mathematical construction with possible applications beyond model (1).

Weaknesses

At first site I found the Hamiltonian (1) somewhat artificial. The coefficient of the non-local term has to be fine tuned for the presented approach to work. However, section 2.3 dispels these doubts to large extent. Solvable models have to be special but 2.3 shows that (1) is close to some generic models.

Report

This is a very imaginative mathematical physics that, however, should be readable to broad audience. I mean that the formalism is relatively straightforward and, therefore, may be applicable in other models as well. The work is certainly worth being published in Scipost Physics.

---

## Round 1 · Referee Report · Anonymous (Referee 2) · 2022-12-13

Strengths

  1. The manuscript shows that the TIFM model with an additional fully-connected interacting term can be solved exactly in the thermodynamic limit, which corresponds to a MF solution. This, in my opinion, is very interesting as contrary to popular believe the local terms do not spoil the MF result. Moreover, there is a possibility that similar argument can also be applied to a broad class of mixed-ranged Hamiltonians.

  2. The analytical treatment provided in the manuscript is very innovative and presented in a detailed and readable manner that can be followed by broad audience of SciPost Physics.

Weaknesses

See report.

Report

In the manuscript, the Author has considered standard 1D TIFM with additional fully-connected interacting term. The manuscript is divided into two parts — (a) in the first part, they have solved the system in the thermodynamic limit by means of ‘density-resolved coherent states’ that correspond to a mean-field Hamiltonian (MFH), essentially proving that MF treatment of the system becomes exact in the thermodynamic limit; (b) in the second part, the Author has then analyzed the system both at zero and finite temperatures using previously found analytical results.

Overall, I believe that the present work should deserve a publication in SciPost Physics in some form. But the Author should first modify the manuscript accordingly to make it suitable for the same. My criticisms/comments are as follows:

  1. The TIFM with all-to-all interaction, although looks artificial, has immense practical importance as similar systems can be realized with ultra-cold atoms on optical lattices interacting with high-finesse cavities. The Author should include a discussion about many-body cavity systems in Sec. 2.3.

  2. The present analytical treatment, although very innovative and interesting, can only work if the short-range part of the Hamiltonian is exactly solvable, making similar analysis in other model with mixed-ranged interaction intractable. The Author should mention this clearly in the manuscript. It would be great, but not mandatory, if the Author can comment whether the MF treatment also becomes exact in the thermodynamic limit for such ‘unsolvable’ systems.

  3. While the first part of the manuscript is almost complete, the second part must be improved to a large extent. To make the manuscript readable for broad audience, the Author should include several illustrative and quantitive plots. Please follow following points for that.

  4. In the table (wrongly denoted as Figure 1) the Author has compared ED energies with the thermodynamic MF result. In my opinion, such comparison should be accompanied by a plot, where they should plot energies of exact and MF Hamiltonian for different $L$, along with their extrapolations (with error-bars), and the exact thermodynamic MF result. Also, the $1/L$ extrapolation with only $L=18, 20$ is very crude and must be improved with proper error analysis. With present-day ED methods, it is possible go upto $L=24 - 28$ using standard computers for this Hamiltonian to do such extrapolation. They may also consider other observables for such comparisons.

  5. The Author should clarify what do they mean by ‘ordered’, ‘disordered’, and ‘paramagnetic’ just before the Sec. 4.2 in terms of proper order parameters.

  6. The left panel of Figure 2 and both the panels in Figure 3 must include proper numerical values of the axis parameters. The temperature is also not quoted in the caption of the left panel of Figure 3.

  7. The Author should show the variations of different order parameters across different phase transitions, both from the numerics (for exact and MF Hamiltonians) for different system sizes, and in the thermodynamic limit using MF results. Since, average magnetizations and average two-site correlators are diagonal in momentum basis, they should be calculable within the present framework.

Requested changes

See report.

---

## Round 2 · Referee Report · Anonymous (Referee 1) · 2023-1-19

Report

I sustain my previous recommendation to publish the manuscript.

Thanks to the detailed suggestions by the second referee its readability has been further improved beyond the original version.

---

## Round 2 · Referee Report · Anonymous (Referee 2) · 2023-2-21

Report

The Author has considered all the points of my previous report to varying extent in the updated manuscript. Therefore, I recommend its publication in SciPost Physics. I apologize to the Author and the Editor for the delay.

---

## Round 2 · Author Response

I thank both referees for their careful reading and positive comments about the draft. In particular I thank the first referee for their very positive comments and recommendation for publication. As for the particular points raised by the second referee, here are my answers and list of changes:

1 - I included a discussion in section 2.3.2 as requested by the referee.

2 - As the referee points out, the present method indeed works only when the short-range Hamiltonian is exactly solvable. Treating generic cases is beyond the scope of this work, but I added in the last paragraph of the conclusion a sentence to say that the exact solvability is required in this paper.

3 - I understand the comment of the referee that adding some plots would make the second part of the manuscript more accessible to a large audience. As detailed below, I added new plots to illustrate the different results. However, I also have to remark that the analytic study in Section 4 of the different phases obtained from the mean field solution of Section 3 is already very detailed, precise and complete.

4 - I added a more precise plot in Section 3.9 as requested by the referee. The obtained relative precision is of order at most $10^{-4}$ using a simple linear fit on the data. The precision and relevance of the linear fit is clearly visible in the plot.

5 - I clarified the meaning of disordered and ordered at the end of 4.1.3 as requested by the referee.

6 - I added numerical values in the left panel of Fig 4, and quoted the temperature used. In the other panels numerical values were already present through the location of the critical points, but I added some additional labels to make it clearer.

7 - I thank the referee for this good suggestion. I added a plot in Fig 3 for e.g. the first order transition in $\sigma^x$ at zero temperature, with comparison between the theory and the numerics.

Best regards

Etienne

---

## Round 2 · List of Changes

See comments.

---

## Editorial Decision

published